# Pilot Feasibility Study of a Multi-View Vision Based Scoring Method for Cervical Dystonia

**DOI:** 10.3390/s22124642

**Published:** 2022-06-20

**Authors:** Chen Ye, Yuhao Xiao, Ruoyu Li, Hongkai Gu, Xinyu Wang, Tianyang Lu, Lingjing Jin

**Affiliations:** 1Department of Computer Science and Technology, Tongji University, 4800 Caoan Road, Shanghai 201800, China; yechen@tongji.edu.cn (C.Y.); 2033089@tongji.edu.cn (Y.X.); 2030801@tongji.edu.cn (X.W.); tylu@alumni.tongji.edu.cn (T.L.); 2The Key Laboratory of Embedded System and Service Computing Ministry of Education, Tongji University, 4800 Caoan Road, Shanghai 201800, China; 3Neurotoxin Research Center of Key Laboratory of Spine and Spinal Cord Injury Repair and Regeneration of Ministry of Education, Neurological Department of Tongji Hospital, School of Medicine, Tongji University, 389 Xincun Road, Shanghai 200065, China; 1910938@tongji.edu.cn (R.L.); 2033217@tongji.edu.cn (H.G.); 4Department of Neurology and Neurological Rehabilitation, Shanghai Yangzhi Rehabilitation Hospital (Shanghai Sunshine Rehabilitation Center), School of Medicine, Tongji University, Shanghai 200092, China

**Keywords:** Azure Kinect, Cervical Dystonia, human motion analysis, human pose estimation, remote diagnosis

## Abstract

Abnormal movement of the head and neck is a typical symptom of Cervical Dystonia (CD). Accurate scoring on the severity scale is of great significance for treatment planning. The traditional scoring method is to use a protractor or contact sensors to calculate the angle of the movement, but this method is time-consuming, and it will interfere with the movement of the patient. In the recent outbreak of the coronavirus disease, the need for remote diagnosis and treatment of CD has become extremely urgent for clinical practice. To solve these problems, we propose a multi-view vision based CD severity scale scoring method, which detects the keypoint positions of the patient from the frontal and lateral images, and finally scores the severity scale by calculating head and neck motion angles. We compared the Toronto Western Spasmodic Torticollis Rating Scale (TWSTRS) subscale scores calculated by our vision based method with the scores calculated by a neurologist trained in dyskinesia. An analysis of the correlation coefficient was then conducted. Intra-class correlation (ICC)(3,1) was used to measure absolute accuracy. Our multi-view vision based CD severity scale scoring method demonstrated sufficient validity and reliability. This low-cost and contactless method provides a new potential tool for remote diagnosis and treatment of CD.

## 1. Introduction

Dystonia was first proposed by Oppenheim [1]. In 2013, dystonia was redefined as dyskinesia with clinical features of abnormal and repetitive movements caused by continuous or intermittent muscle contraction [2]. Cervical Dystonia (CD) is the most common focal dystonia in the clinic [3]. It is dystonia caused by clonic or tonic excessive contraction of cervical muscles, which leads to abnormal head and neck posture and involuntary movement. Common treatment methods include drug therapy, botulinum toxin injection therapy, and surgical treatment. Local injection of botulinum toxin can effectively reduce muscle contraction and pain, which is recommended as the first treatment by the American Neurological Association and the European Union of neuroscience associations [4,5]. The effective rate of this clinical treatment method is 58–90% [6]. The main reason for treatment failure is the insufficient judgment of the responsible muscles [7].

In clinical practice, the target muscle of botulinum toxin therapy is usually selected by the abnormal movement pattern of the patient. According to the clinical manifestations, the abnormal movement patterns of patients can be classified as torticollis (rotation in the transverse plane), laterocollis (lateroflexion in the coronal plane), retrocollis (anteflexion in the sagittal plane), anterocollis (retroextension in the sagittal plane), or a combination of the above [8]. With the widespread use of botulinum toxin therapy, researchers found that the past four abnormal movement patterns had described cervical dystonia too simply to account for the various clinical symptoms, making it difficult to select the responsible muscle for treatment. According to the collum-caput concept, each pattern can be divided into two subtypes. One is abnormal movement relative to the neck from the head, and the other is abnormal movement relative to the neck from the torso. Each abnormal movement pattern corresponds to a part of the muscles [9,10,11,12]. As shown in Figure 1, torticollis can be subdivided into torticollis and torticaput. Laterocollis can be subdivided into laterocollis and laterocaput. Antecollis can be subdivided into antecollis and antecaput. Retrocollis can be subdivided into retrocollis and retrocaput.

### 1.1. Related Work

Accurate scoring of the severity scale is significant for botulinum toxin therapy. There are three methods commonly used in the clinical scoring of the severity scale. The manual assessment uses a head protractor [13] to assess abnormal patterns of CD. Each patient needs to put the head and neck into the instrument in this method. The doctor uses the protractor to manually measure the patient’s head and neck movement angle under different states. This method is complex, time-consuming, and exhausting.

The Inertial Measurement Unit (IMU) based method uses a multi-axis sensor combination device [14] to assess abnormal patterns of CD. Doctors attach IMUs to the patient’s neck and head to measure the motion angles. Compared with the manual assessment method, this method is more convenient. However, physical contact will make the patient uncomfortable. The patient needs to wear a head positioning cap [15] to ensure that the sensors are correctly positioned, as shown in Figure 2. The head positioning cap is tightly tied to the head, which will make the patient feel uncomfortable or even painful, especially for obese people. The drifting problem is that the inherent imperfections and noise within the IMU will cause errors. These errors will accumulate as angle estimates drift over time. It will result in a difference between the estimated and actual angles, which will cause errors in the measurement.

X-ray assessment accurately determines the abnormal movement patterns of dystonia by comparing the positions of different vertebral bodies and the cross-sectional area of the corresponding muscles in the neck through multi-angle X-ray photography of the head and spine. This method is more accurate than manual assessment and sensor assessment, but it increases patient radiation exposure, and the operation is more complex. Conventional Computed Tomography (CT) requires the patient to be supine, which may interfere with the patient’s abnormal posture. Therefore, the assessment of abnormal patterns of CD needs to be completed under the special orthostatic CT.

In recent years, vision based human pose estimation (HPE) has made significant progress and has begun to be widely used in related fields. Two-dimension (2D) HPE methods such as Convolutional Posture Machine (CPM) [16], Hourglass [17], and High-Resolution Net (HRNet) [18] estimate the 2D position of human body keypoints from images or videos. Three-dimension (3D) HPE methods such as SimpleBaseline3D [19] and VideoPose3D [20] estimate the 3D position of human body keypoints. It is possible to use computer vision technology in medical diagnosis. Li et al. used the CPM to extract the joint motion trajectory from the video and then calculated the relevant characteristic parameters. The patient’s pathological motion was measured using the random forest to predict their clinical class [21]. Guo et al. extracted keypoints from the patient’s video to assess the abnormal level of Parkinson’s disease [22]. Viswakumar et al. proposed a cost-effective method for human gait analysis that uses a mobile phone camera [23]. Nakamura et al. developed a system analyzing the three-dimensional keypoint positions obtained by Kinect v2 [24] to measure head and neck movement angles. It can calculate the TWSTRS severity scale score semi-automatically [25]. Their results showed a good correlation between the system and a neurologist in torticollis and laterocollis, but poor agreement in antecollis/retrocollis. Since pitch errors were always higher than yaw and roll errors when using monocular Kinect v2. These cases fully demonstrate the feasibility of vision-based methods in scoring on the severity scale.

### 1.2. Motivation

Above traditional methods requires experienced neurologist trained in dyskinesia and professional medical equipment. In many areas where medical resources are scarce, it is difficult for CD patients in these areas to obtain effective diagnoses. The recent outbreak of Corona Virus Disease 2019 exacerbates the shortage of medical resources. Many CD patients cannot go to the hospital for diagnosis and treatment, which delays the best opportunity for rehabilitation. Therefore, it is urgent to study a low-cost and portable method to score CD scales without a neurologist trained in dyskinesia and professional medical equipment and even realize remote diagnosis at home.

### 1.3. Challenge

One of the challenges of vision based methods is how to accurately describe the head and neck movement of each CD abnormal pattern using the vision based HPE, since the abnormal movement pattern of CD is complex and changeable. Another challenge is the deep ambiguity in estimating 3D body keypoints from a single 2D image. Using multiple views to estimate 3D human keypoints is a preferred solution. Another preferred solution to solve the depth ambiguity problem is using a depth sensor. Yu et al. proposed DoubleFusion [26] using a single depth sensor to estimate 3D human pose. Kadkhodamohammadi et al. used RGB-D sensors to estimate 3D human pose in real operating room environments [27]. Microsoft also released the Kinect body tracking Software Development Kit (SDK) [28] for human 3D keypoint estimation using Kinect’s RGB-D sensors.

### 1.4. Contribution

This study aimed to study a low-cost and portable vision based method for scoring CD scales. The main contributions of this study can be summarized as follows:We propose a multi-view vision based method for scoring the CD severity scale. It measures head and neck movement angle by calculating the 3D keypoint positions obtained by frontal Azure Kinect and the 2D keypoint positions obtained by the common lateral camera.We conducted a pilot study to compare the subscales of the Toronto Western Spasmodic Torticollis Rating Scale (TWSTRS) severity scale calculated by the multi-view vision based method with the manual method scores rated by a neurologist trained in dyskinesia. The results show a good correlation and agreement between the two methods. It demonstrated sufficient validity and reliability of the multi-view vision based method.We compare the subscales of the TWSTRS calculated by the multi-view vision based method with the scores rated by the single-view method and wearable IMU based method. The results show that our multi-view vision based method has higher accuracy and robustness than the single-view method and wearable IMU based method.

Compared with the traditional scoring methods, our vision-based method uses the patient’s image to assess the abnormal movement pattern. This method only uses the camera device to collect image data without direct contact with the patient. While maintaining the accuracy of the assessment, it is convenient, quick and easy to be applied in remote diagnosis and treatment. Compared with the single view methods, our multi-view method has significantly improved accuracy at the cost of adding a lateral camera.

The work is structured as follows: In Section 2, we describe the subjects, proposed methods, and devices in detail. In Section 3, we show the superiority of our method. The discussion is presented in Section 4. Section 5 summarizes the article.

## 2. Methods

### 2.1. Subject

The Institutional Review Board of Tongji Hospital, Tongji University School of Medicine approved this pilot study. This study was conducted based on the clinical videos from the Neurology Department of Tongji Hospital. There are now 31 patients participating in our pilot study. Of these, 8 participants have completed data collection and 23 others are waiting for data collection. The data contains 24 frontal videos, 24 lateral videos and 24 IMU data of 8 CD patients. The movement disorder-trained neurologist collected detailed participant information and scored the TWSTRS severity scale. Frontal videos were captured using an Azure Kinect [28] at 30 frames per second at a resolution of 1920 × 1080. Lateral videos were captured using the HP 320 FHD Webcam [29] at 30 frames per second at a resolution of 1920 × 1080. IMU data was synchronously captured using three Alubi LPMS-B2 [30] inertial measurement units at 400 Hz. The participants were seated or standing and facing the Azure Kinect in all videos.

### 2.2. Multi-View Vision Based Method

The multi-view vision based scoring method uses the Azure Kinect to capture the frontal color and Infrared Radiation (IR) images and the HP 320 FHD Webcam to get the lateral color image, as shown in Figure 3. The Azure Kinect is a new RGB-D sensor released by Microsoft. It consists of an RGB camera and an IR camera. The resolution of the color camera is 1920 × 1080 px at 30 fps. The resolution of the IR camera is 512 × 512 px at 30 fps. Its weight is 440 g, and its size is 103 mm × 39 mm × 126 mm [28]. The HP 320 FHD Webcam is a USB camera released by Hewlett-Packard. It has a resolution of 1980 × 1080 px at 30 fps. Its weight is 130 g, and its size is 72 mm × 54 mm × 23 mm [29]. The frontal Azure Kinect was placed at a distance of around 2.0 m in front of the subject, while the HP 320 FHD Webcam was placed at a distance of about 2.0 m on the side of the subject. The Azure Kinect Body tracking SDK captures the 3D keypoint positions of the subject, and the later 2D keypoint positions of the subject are captured by the You Only Look Once (YOLO) model [31,32] and HRNet [18] model. The frontal 3D keypoint positions are used to automatically calculate the yaw axis angles (rotation) and roll axis angles (lateral tilting). The 2D lateral position is used to automatically calculate the pitch axis angles (sagittal flexion and extension).

The flow chart of our method is shown in Figure 4. The Azure Kinect Body tracking SDK consists of 2D pose estimation and 3D model fitting. In the 2D pose estimation module, the convolutional neural network is used to extract features from the RGB image of the color camera and obtain the linked 2D keypoints. The 3D model fitting module uses depth images from the IR camera and linked keypoints from the 2D pose estimation module as input. Energy data terms (include 2D keypoint reprojection and 3D surface depth displacement) and energy regularization terms (include anatomical joint limits, pose prior regularization, scale prior regularization, and temporal coherency) are used to fit and optimize the kinematic pose of the subject. Each body pose comprises 32 joints (head, neck, nose, etc.), each characterized by a 2D keypoint position.

The keypoints of the subject used in our method are shown in Figure 5. And the scheme of head and neck movement angle calculation is shown in Figure 6. With the frontal 3D keypoint positions of the subject, the torticaput/torticollis and latercaput/laterocollis subscales of the TWSTRS severity scale can be scored. For torticaput, the angle between the vector passing through both ears and the horizontal vector in the transverse plane is calculated. For torticollis, the angle between the neck-head vector and the horizontal vector in the transverse plane is calculated. For latercaput, the angle between the vector passing through both eyes and the horizontal vector in the coronal plane is calculated. For laterocollis, the angle between the neck-head vector and the vertical vector in the coronal plane is calculated.

The YOLOv3 model, which performs well in the subject detection task, is used to obtain subject bounding boxes in the lateral image. The YOLOv3 model is a single-stage object detection model, which can maintain high detection accuracy while having a high detection speed, and occupies fewer computing resources. It is suitable for real-time medical diagnosis. To improve the accuracy of the YOLOv3 model for subject detection, we pre-trained the YOLOv3 model with the Common Objects in Context (COCO) [33] dataset.

The HRNet model is used to estimate the subject’s 2D keypoint positions. The backbone network of HRNet has maintained high resolution rather than recovery from the low-resolution features. Through up-sampling and down-sampling with mutual connection, the parallel network realizes the multi-scale feature extraction and fusion. It solved the need for a reliable high-resolution feature map in the task of human keypoint estimation.

With the lateral 2D keypoint positions of the subject, the antecaput/retrocaput and antecollis/retrocollis subscale of the TWSTRS severity scale can be scored. For antecaput/retrocaput, the angle between the vector passing ear-nose and the horizontal vector in the sagittal plane is calculated. For antecollis/retrocollis, the angle between the vector passing neck-ear and the vertical vector in the sagittal plane is calculated.

### 2.3. Single-View Vision Based Method

Both multi-view and single-view methods use the same human keypoints to calculate the motion angle. So the scheme of using human keypoints to calculate the head and neck motion angle is the same, as shown in Figure 6. The difference between multi-view and single-view methods is that the 2D lateral human keypoints are obtained differently. The single-view method without the lateral camera firstly estimates 3D human keypoints using the Azure Kinect. It then projects the 3D keypoints to the sagittal plane to get 2D lateral keypoints. The multi-view method directly estimates 2D lateral keypoints from the lateral camera.

### 2.4. Manual Measurement

Manual measurement is the method used by the neurologist trained in dyskinesia to score CD scales. This method uses the protractor to measure the neck and head motion angle. For antecaput/retrocaput, the angle between the vector from the external auditory foramen to CZ (the top of the head) and the vertical vector in the sagittal plane is calculated. For antecollis/retrocollis, the angle between the vector passing through the midpoint of the neck and parallel to the neck contour and the vertical vector in the sagittal plane is calculated. For latercaput, the angle between the vector from the nose to the glabellum and the vertical vector in the coronal plane is calculated. For laterocollis, the angle between the vector from the midpoint of the thyroid cartilage to the suprasternal notch and the vertical vector in the coronal plane is calculated. For torticaput, the angle between the vector from nose to external occipital protuberance and the horizontal vector in the sagittal plane is calculated. For torticollis, the angle between the vector from the midpoint of the thyroid cartilage to the C4 spinous process and the horizontal vector in the transverse plane is calculated.

### 2.5. IMU Based Method

The IMU based method captures the neck and head motion angle from 3 LPMS-B2 attached to the subject body. The LPMS-B2 is a wearable inertial sensor released by Alubi. It can capture 6 degrees of freedom data from a triaxial accelerometer and a gyroscope at 400 Hz. Its weight is 12 g, and its size is 39 mm × 39 mm × 8 mm [30]. As shown in Figure 7, the S1 sensor is attached to the top of the head, the S2 sensor is attached to the C3-C4 of the cervical spine, and the S3 sensor is attached to the C7 of the cervical spine. These sensors can measure the angle of pitch, roll, and yaw using three rate gyros. For torticaput, the angle of yaw between S1 and S2 is calculated. For torticollis, the angle of yaw between S2 and S3 is calculated. For latercaput, the angle of roll between S1 and S2 is calculated. For laterocollis, the angle of roll between S2 and S3 is calculated. For antecaput/retrocaput, the angle of pitch between S1 and S2 is calculated. For antecollis/retrocollis, the angle of pitch between S2 and S3 is calculated.

### 2.6. Study Protocol

In this study, three methods were used to score the rotation (torticaput/torticollis), laterocollis (latercaput), and antecollis/retrocollis (antecaput/retrocaput) subscales of the TWSTRS severity scale, which are the multi-view vision based method, neurologist trained in movement disorders and the wearable IMU based method. The antecollis/retrocollis subscale was also scored by the single-view vision-based method, which calculated angles only using the frontal keypoint positions captured by the frontal Azure Kinect.

### 2.7. Data Analysis

Data were analyzed to ascertain the coefficient of correlation between the subscales of the TWSTRS calculated by the multi-view vision based method and the scores obtained by a movement disorder-trained neurologist. The coefficient of correlation between the subscales of the TWSTRS was calculated by the sensor based method, and the scores obtained by a movement disorder-trained neurologist were also ascertained. Spearman’s correlation was used to evaluate the relative agreement between these methods. Intra-class correlation (ICC) and 95% limits of agreement were used to measure absolute accuracy. The SciPy library v1.7.3 was used to calculate Spearman’s correlation, and the Pingouin library v0.5.1 was used to calculate ICC.

## 3. Results

Table 1 shows the characteristics of the subjects. Eight participants had completed data collection in this study. The male to female ratio is 3:5, and the average age was 41.3 years. These patients were diagnosed and treated in the Neurological Department of Tongji Hospital.

Table 2 shows the angles and the subscales of the TWSTRS severity scale calculated by the multi-view vision based method, as well as those calculated by a neurologist trained in movement disorders and the wearable inertial sensors based method. For the antecollis/retrocollis subscale, Table 2 also shows the severity score calculated by the single-view vision base method, which only uses the frontal image from the frontal Azure Kinect.

As shown in Table 3, the scores calculated by our method were significantly correlated with those measured by the neurologist (r = 0.843 for rotation, r = 0.667 for laterocollis, and r = 0.701 for antecollis/retrocollis). Adequate ICC(3,1) of 0.870(*p* < 0.05) was obtained in rotation, 0.727(*p* < 0.05) was obtained in laterocollis, and 0.739(*p* < 0.05) was obtained in antecollis/retrocollis. We compared our method’s validity and accuracy with those of Nakamura’s method [25], which uses a single Kinect v2 to capture the three-dimensional position of the subject. As we can see, our method has a significant improvement in validity and accuracy in laterocollis (r = 0.369, icc = 0.330 by Nakamura) and antecollis/retrocollis (r = 0.181, icc = 0.281 by Nakamura).

In this study, we also compared the validity and accuracy of the multi-view vision based method with the wearable IMU based method. Spearman’s correlation coefficient and ICC(3,1) between the scores was calculated by the wearable IMU based method and neurologist are shown in Table 4. It is shown that both the validity and accuracy of the vision based method are higher than the IMU based method in all subscales.

As the ablation experiment, Spearman’s correlation coefficient and ICC(3,1) between the score of antecollis/retrocollis subscale measured by the single-view vision based method that only uses frontal image and neurologist were ascertained to study the necessity of adding the lateral image of the subject. As shown in Table 5, with the method of adding the lateral image of the subject, there was a significant improvement in validity and accuracy compared to the method that only uses the frontal image.

In addition, we counted the actual costs of implementation and maintenance of the above methods, as shown in Table 6. The total cost of devices used in our multi-view vision based method is 1518 USD, while the manual measurement method is 100 USD and the IMU based method is 1589 USD.

## 4. Discussion

Our results are comparable with a previous study that used a single Kinect v2 to capture the 3D keypoint positions of the subject [25]. As shown in Table 2, our multi-view vision based method has a significant improvement compared with the previous work. The validity and accuracy of the vision based method strongly depends on the precision and accuracy of the skeleton tracking (human keypoint estimation). There are previous studies that evaluated the Azure Kinect and its discontinued predecessors, which found that the Azure Kinect surpasses the Kinect v2 both in precision and accuracy [34,35].

The Azure Kinect includes a higher resolution depth sensor and deep learning-based body tracking algorithms; a convolutional neural network model might provide more accurate kinematic measurements than Kinect v2, which uses the random forest model [36]. The Body Tracking SDK of the Azure Kinect is able to track 32 joints for each subject, while that of the Kinect v2 only can track 25 joints. In contrast to the skeleton definition of the Kinect v2, the Azure Kinect’s skeleton definition includes more joints in the head, such as ears and eyes. It can help calculate the head movement angle more precisely.

The previous study showed poor agreement in antecollis/retrocollis, since pitch errors are always higher than yaw and roll errors when using Kinect v2 [25]. Clark et al. also found that it would be more prone to measurement errors when tracking the body from the frontal view, as it relies solely on the depth measurement in the Z-axis [37]. The Azure Kinect also has the same problem as the Kinect v2. Our results showed mediocre agreement in antecollis/retrocollis when using the single frontal Azure Kinect. It has proved the necessity of adding the lateral view in the vision base method. More precise human keypoint positions in the pitch axis can be captured by deep learning based pose estimation algorithms such as HRNet using the lateral view image.

Compared with the vision based method, the wearable IMU based method has mediocre validity and accuracy. The physical contact between the sensor and the patient’s body will inevitably interfere with the patient’s movement, and the patient will feel uncomfortable. The wearable inertial sensors are attached to the subject’s skin, but the position of the sensor will deviate each time, and the muscle movement state and the skin movement state will be inconsistent. The drifting problem of the IMU also might cause errors. All these factors will cause errors in the measurement.

The device cost of vision based method is higher than manual measurement. But the medical resources cost of vision based method is significantly lower than manual measurement. Our vision based method can automatically assess abnormal movement patterns without a neurologist trained in dyskinesia. Compared with a sensor based method, the device cost for the vision based method is slightly lower. Moreover, the sensor based method also requires a doctor to attach the sensors to the correct position on the patient’s body.

The Azure Kinect has a similar performance to the Vicon Motion System, another body tracking system in real-life scenarios [36]. Compared with the Vicon Motion System, the Azure Kinect does not require the time-consuming placement of markers, and it is portable and cost-effective. The method using the Kinect could easily reach more patients. It is also easy to be applied in remote diagnosis and treatment for patients living in remote areas who do not have access to medical services.

### 4.1. Limitations

This study has some limitations that are important to be aware of. Firstly, the abnormal pattern of CD is complex and diverse, which can be combined by different base abnormal patterns. For example, some patients might have symptoms of both antecollis and laterocollis. However, there were only 8 participants enrolled in this study, which is not sufficient to cover the whole breadth of abnormal patterns. It might limit the universality of these results to the majority of patients. Furthermore, our current method calculates the motion angle separately for each abnormal pattern and then scores the scale without considering the correlation between each abnormal pattern. In fact, for most of the obtained patient’s diagnosis reports, there is a certain correlation between abnormal patterns. For example, laterocollis is often accompanied by latercaput, and antecollis is often accompanied by antecaput. The symptoms of one abnormal pattern might affect the angle calculation of another abnormal pattern, which might cause errors.

Different from Parkinson’s disease, abnormal level assessment and gait analysis, which just use the human skeleton for analysis, the scoring of the CD severity scale required detailed position information of the human head and neck. However, the existing keypoints of human datasets such as COCO and Human3.6 M [38] are used for human action recognition, behavior analysis, human-computer interaction, and other tasks. These datasets focus on the human body parts, while the description of the head and neck is too simple to assess the abnormal patterns of CD. For example, there is no keypoint about the neck in the COCO dataset. It leads to the conclusion that the movement of the head and neck cannot be precisely depicted by such keypoints of the patient. The 3D human keypoint dataset Human3.6 M only has one keypoint describing the entire human head. It also cannot accurately describe the movement detail between the head and neck, which might lead to a low accuracy rate in the assessment of abnormal patterns about the neck. Although Azure Kinect SDK can track more keypoints about the human head, such as the eyes and ears, it is not sufficient to precisely describe the abnormal movement of CD. For example, a neurologist trained in dyskinesia usually uses special human parts such as the C4 spinous process, thyroid cartilage and suprasternal notch to measure the neck and head motion angle. However, there are no datasets that contain these special human keypoints.

There is another limitation that must be mentioned. Although the vision based method using Azure Kinect is lower cost and more efficient than the traditional method, the Azure Kinect Body tracking SDK has stringent computer host hardware requirements. The recommended minimum Azure Kinect Body Tracking SDK configuration includes Intel Core i5 Processor and NVIDIA GEFORCE GTX 1050 or equivalent. As shown in Table 6, the cost of these hardware devices is not cheap for some people. It is an extra burden for CD patients with remote diagnosis and treatment at home.

### 4.2. Future Work

In future work, more and more CD patients will gradually participate in this study. With continuous patient data collection, when data reaches a large number, the machine learning methods, which need a lot of patient data to train the model, can be considered to study the correlation between base abnormal patterns.

Artificial intelligence technology such as deep learning has been widely applied in intelligent medicine. To further promote the development of automatic diagnosis and telemedicine based on computer vision, future work could start with building a dataset suitable for use in medicine. The dataset can describe more special keypoints of humans. In this case of the scoring scale of CD, the dataset for neck and head motion analysis containing special human keypoints, such as spinous processes, should be built.

In order to make intelligent diagnoses benefit more patients, future work also could focus on studying how to reduce the computational complexity and memory consumption of the current method. That will be a lightweight deep learning vision based method that can run on common smartphones.

## 5. Summary

This study proposes a multi-view vision based CD severity scale scoring method. This method uses the Azure Kinect to capture the 3D keypoint positions and a common camera to capture the lateral 2D keypoint positions. Finally, it calculates the head and neck motion angles to score the TWSTRS severity scale. The experiments showed a good correlation between our method and the movement disorder-trained neurologist. It demonstrates the validity and accuracy of our method. Compared with the traditional scoring method, our multi-view vision based method is contactless, portable and cost-effective. It provides a new potential tool for clinical diagnosis.

## Figures and Tables

**Figure 1 sensors-22-04642-f001:**
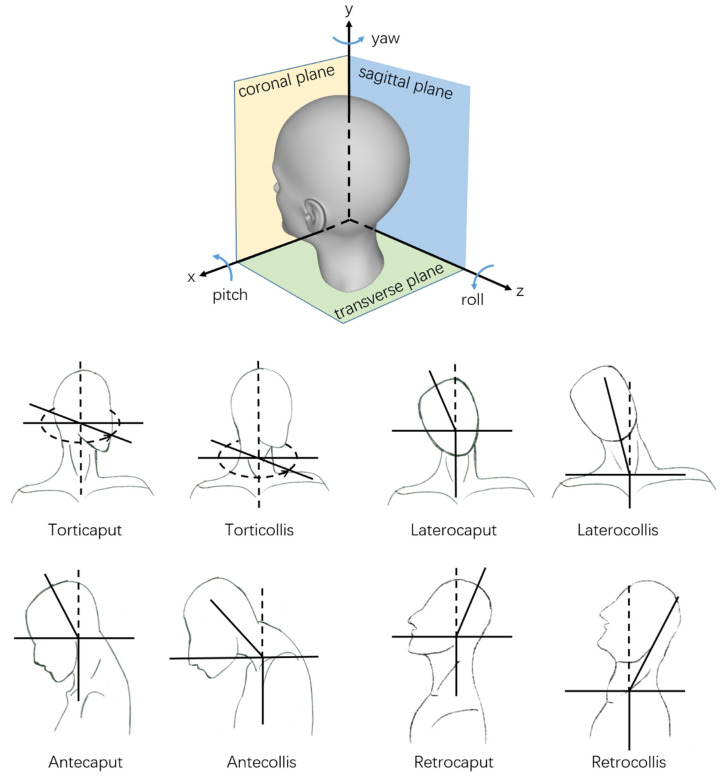
The diagram of abnormal movement patterns of CD.

**Figure 2 sensors-22-04642-f002:**
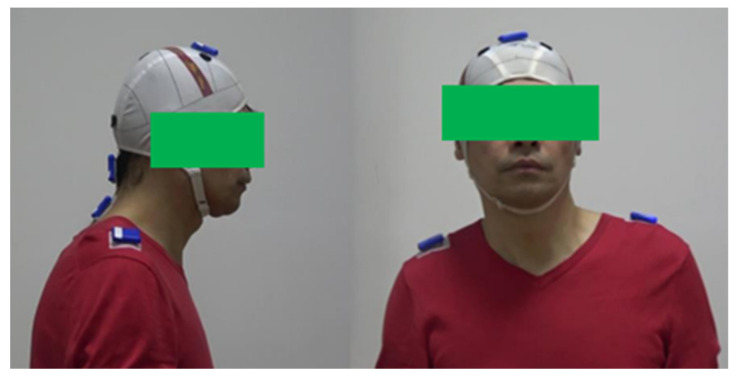
Head positioning cap.

**Figure 3 sensors-22-04642-f003:**
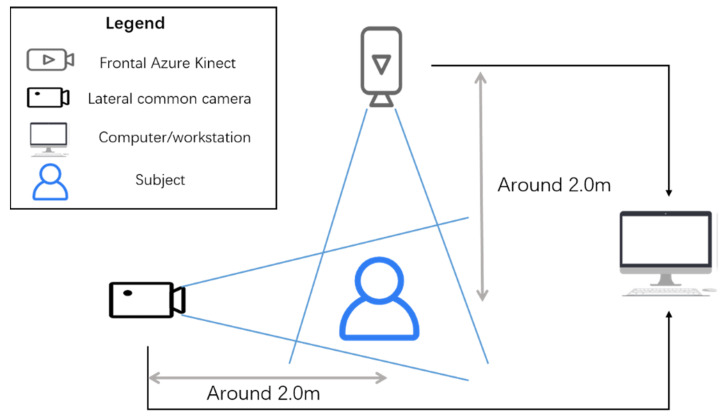
Devices Diagram for the multi-view vision based method.

**Figure 4 sensors-22-04642-f004:**
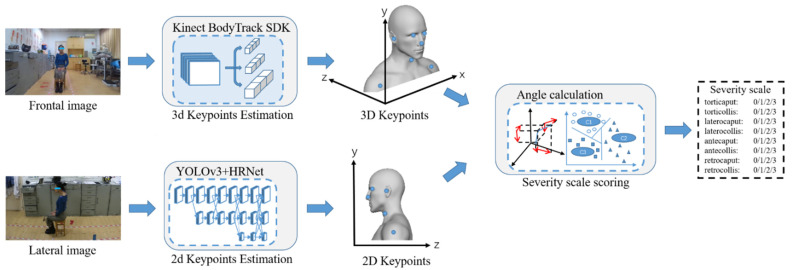
The multi-view vision based method.

**Figure 5 sensors-22-04642-f005:**
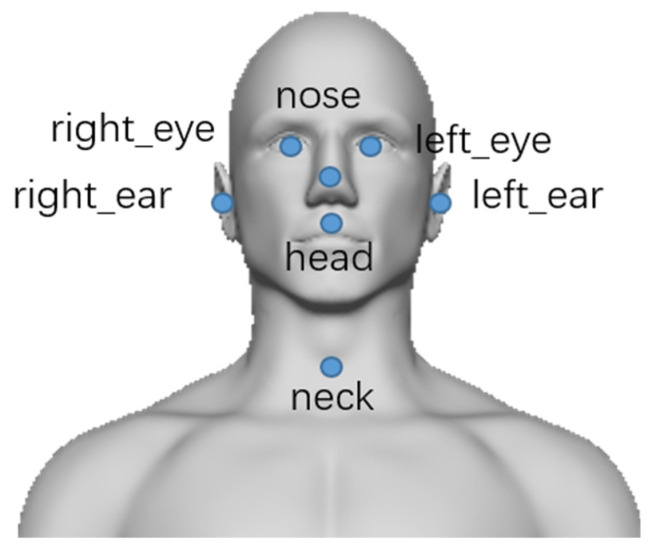
The human keypoints of the subject.

**Figure 6 sensors-22-04642-f006:**
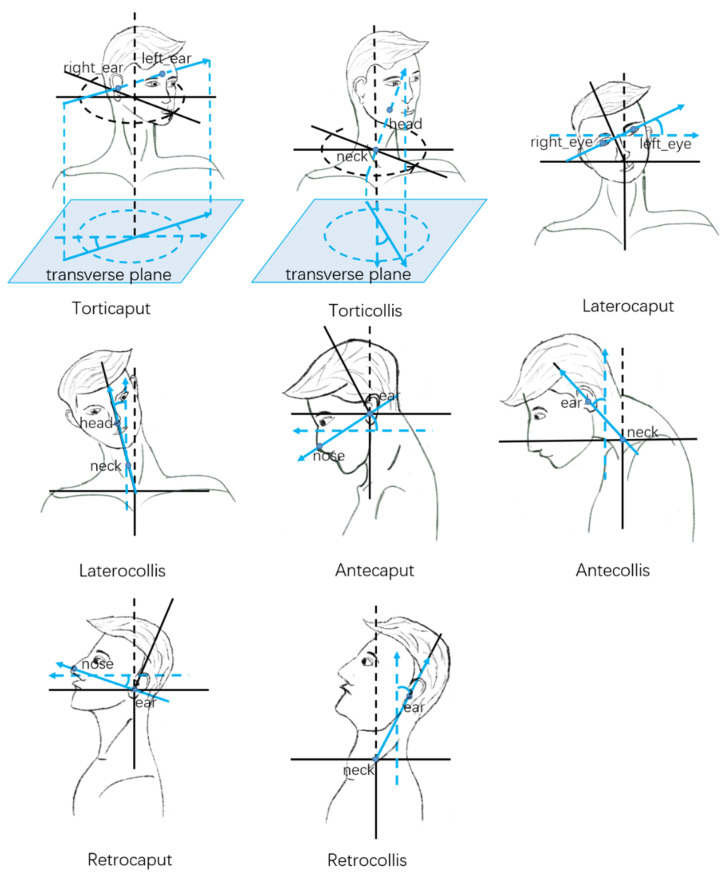
The scheme of angle calculation.

**Figure 7 sensors-22-04642-f007:**
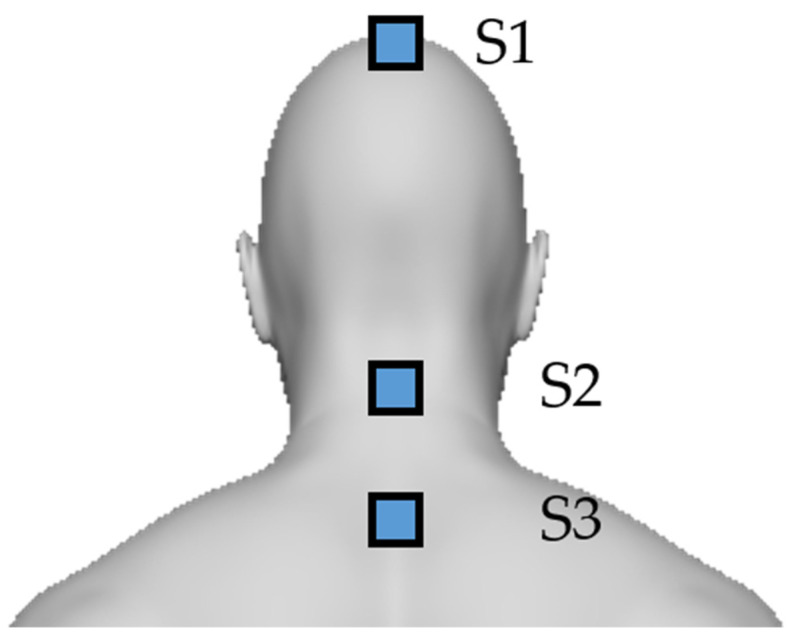
The scheme of IMU based method.

**Table 1 sensors-22-04642-t001:** Characteristics of subjects.

Subject	Age (Years)	Sex	TWSTRS Subscales		
			Rotation	Laterocollis	Antecollis/Retrocollis
1	26	Female	Mild	Moderate	Moderate
2	48	Female	Severe	None	None
3	45	Female	Mild	Mild	None
4	34	Female	Slight	Moderate	None
5	52	Female	Slight	Moderate	Mild
6	37	male	Slight	Moderate	Mild
7	46	male	Slight	Mild	Severe
8	42	male	Mild	Moderate	Moderate

**Table 2 sensors-22-04642-t002:** Score of the subscales of the TWSTRS severity scale calculated by the multi-view vision based method, a neurologist trained in movement disorders, and the wearable inertial sensors based method.

Patient	Rotation *				Laterocollis *				Antecollis/Retrocollis **				
	RA	N	M	W	RA	N	M	W	RA	N	M	W	F
1	31.67	2	2	2	−34.44	2	2	3	23.20	2	2	1	1
2	48.10	4	3	2	2.30	0	0	1	26.57	0	2	1	0
3	−23.09	2	2	2	−8.16	1	1	1	18.43	0	1	1	1
4	17.85	1	1	1	−11.37	2	1	0	3.94	0	0	0	1
5	6.57	1	1	1	−13.86	2	1	1	20.56	1	1	0	0
6	9.76	1	1	2	−13.20	2	1	2	8.13	1	1	0	1
7	−17.04	1	1	0	−7.85	1	1	1	−90.00	3	3	2	2
8	15.56	2	1	1	−28.87	2	2	1	−45.00	2	2	1	1

RA: raw angle by the multi-view vision based method, N: movement disorder-trained neurologist, M: multi-view vision based method, W: wearable IMU based method, F: single-view based method only using the frontal image. * Negative value represent right. ** Negative value represent posterior.

**Table 3 sensors-22-04642-t003:** Validity and accuracy of our method and the previous work.

Items		Correlation	ICC(3,1)
Rotation	Nakamura’s	0.902 *	0.793 *
	Ours	0.843 *	0.870 *
Laterocollis	Nakamura’s	0.369 *	0.330 *
	Ours	0.667	0.727 *
Antecollis/retrocollis	Nakamura’s	0.181	0.281
	Ours	0.701	0.739 *

* *p* < 0.05.

**Table 4 sensors-22-04642-t004:** Validity and accuracy of the multi-view based method and the wearable IMU based method.

Items		Correlation	ICC(3,1)
Rotation	M	0.843 *	0.870 *
	W	0.564	0.484
Laterocollis	M	0.667	0.727 *
	W	0.189	0.211
Antecollis/retrocollis	M	0.701	0.739 *
	W	0.474	0.525

M: multi-view vision based method, W: wearable IMU based method. * *p* < 0.05.

**Table 5 sensors-22-04642-t005:** Validity and accuracy of the multi-view vision based method and the single-view vision based method.

Items		Correlation	ICC(3,1)
Antecollis/retrocollis	M	0.701	0.739 *
	F	0.550	0.532

M: multi-view vision based method, F: single-view vision based method only using the frontal image. * *p* < 0.05.

**Table 6 sensors-22-04642-t006:** The costs of implementation and maintenance of the methods.

Method	Device	Price (USD)	Total Price (USD)
Vision based method	Azure Kinect	399	1518
	HP 320 FHD Webcam	29	
	Computer (include graphic processor)	1090	
Manual Measurement	Professional protractor	100	100
IMU based method	LPMS-B2	180 × 3	1589
	Computer	990	
	Head position cap	59	

## Data Availability

The data that support the findings of the study are available from Tongji Hospital Affiliated to Medical College of Tongji University but restrictions apply to the availability of these data, which were used under license for the current study, and so are not publicly available. Data are however available from the authors upon reasonable request and with permission of Tongji Hospital Affiliated to Medical Collage of Tongji University.

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
