# Peer review of "Pilot Feasibility Study of a Multi-View Vision Based Scoring Method for Cervical Dystonia"

_sensors, 2022, doi:10.3390/s22124642_

Round 1

Reviewer 1 Report

The research topic is interesting and the work has potential; however, the paper has many shortcomings in regards to validation, the data collection, discussion, clarity and presentation. The recommendation is for rigorous revision. Below are some comments.

Some ideas in the first paragraph (botulinum toxin treatment recommendations and the reason for treatment failure) should have citations supporting them. An explanation on why the sensor measurement contact is uncomfortable for the patients should be included (line 70). On this topic, it is unclear what the drifting problem is and how it causes measurement errors. The type of sensors are mentioned only at the end of the introduction (inertial), but it is unclear if they are the same sensors described in current methods. The introduction requires some revision to showcase the limitations of current methods and clarify the concepts, ideas, and arguments.

The papers cited in related works are refered to by the first author only; instead, if multiple authors, the text should reference them as: both author surnames if there are 2 authors; first author surname "et al." if there are more than two. The lists of datasets in Sect. 2.4 end with "et al." and since this means "and the others", perhaps they should be named or at least a reference to a list should be provided. There should be references to at least the websites of the sensors and Kinect systems discussed / used in the paper.  

Acronyms are used throughout the paper without explicating their meaning at first use. The abstract does not count, so the suggestion is to begin explicating acronyms in the introduction for enhanced readability. Acronyms like 2D and 3D should be written with capital D.

Subjects without cervical dystonia were not included. It is not described how validity is assessed. It is only described that validity and accuracy are compared against another Kinect-based method, but no validation against reliable clinically validated methods is described. The other Kinect-based study the proposed method is compared against is incorrectly referenced and it uses a different dataset. There are not enough details provided on the subjects that could affect the measurements.

The study claims cost effectiveness, but neither the Azure Kinect (~400 USD), nor its maintenance (hardware or software) and other hardware dependencies such as specialized graphic processors, are cheap. The claims are made against the "traditional method" without providing data regarding actual real costs of implementation and maintenance.

General formatting and grammar/language revision are needed. For instance the formatting of acronymic names such as "COVID-19", run-on sentences that should be split into separate sentences, incompatible declensions, etc. Title "Human Keypoints Estimation in Medical" of Sect. 2.4 appears to be missing a word. Sect. 3.6 title says "stuty" instead of probably "study". Some sentences do not make sense, e.g. "manual assessment uses [...] to assess" line 63; the sentence starting with "since" at line 164 seems to be missing the main clause; they are not the only ones, so everything should be checked thoroughly.

Reviewer 2 Report

In the work, the authors presented a multi-view video-based cervical dystonia severity scale scoring method. It detected the key points position of the users from the frontal and lateral images, and scored the severity scale by calculating head and neck motion angles. The experiments showed a relatively good correlation between the method and the movement disorder-trained neurologist. Some suggestions are as follows: 

 (1) The contents in Figure 1 and Figure 6 are similar. It is suggested to revise them.

 (2) The background introduction is too much. Please combine Introduction, Related Work, with Motivations and challenges into one part.

 (3) What are the differences between this work and other work, such as methods, advantages, and disadvantages of performance? The authors should clearly point out the innovation of this work.

 (4) Please provide the detailed type, specification, and manufacturer of involved materials and devices.

Reviewer 3 Report

The authors propose a multicamera system that measures head and neck movement angle by calculating the 3D keypoints position obtained by frontal Azure Kinect and the 2D key-points position obtained
by a lateral common camera. The authors also conducted a pilot study to compare the subscales of the TWSTRS severity scale calculated by the multi-view vision-based method with the manual method scores rated by a neurologist trained in dyskinesia.

The article requires a few substantial revisions that are detailed below.

Lines 39-42: Local injection of botulinum toxin can effectively reduce muscle contraction 39
and pain, which is recommended as the first treatment by the American Neurological 40
Association (ANA) and the European Union of neuroscience associations (EFNS). The 41
clinical effective rate of this treatment method is 58%-90% [3].

Are there any statistics on these treatments in China? That would be very interesting to
know. In general, scientific research, especially in the domain of medical sensors, should
broaden the scope of evaluations to go beyond the US/EU. There is a lot of fascinating
medical sensor research in Brazil, India, Russia, and Iran (to name just a few places outside
of the US/EU). The information on these studies in English may be scarce, but every effort
should be made to bring this valuable information into the discussion.

Figure 1 was very helpful to me in understanding abnormal motion patterns on p. 2.

Line 64-65: The patient needs to put the head and neck into the instrument, and the doctor
uses the protractor to manually measure the angle of the patient's head and neck movement
under different states. This method is complex, time-consuming and exhausting.

Yes, this is complex but does not require the deployment of cameras in the patient's home.

Line 67: Sensor measurement uses a multi-axis sensor combination device [10] to assess.

The above statement is not grammatically correct. To assess what?

The introduction section should contain an overview of how the paper is structured. In
Section 2, we review the related work. In Section 3, ..., etc.

Line 143: "2d" must be changed to "2D" and used consistently throughout the paper. For example,
line 162 uses "2D."

Line 145: "3d" must be changed to "3D" and used consistently throughout the paper. For example,
line 161 uses "3D."

Line 174-175: Microsoft also releases the Kinect body tracking SDK for human 3D keypoints estimation
using their RGB-D sensors named Kinect.

I am not familiar with this SDK. Does it consist of the sensors placed on the human body?

Figure 4 was also helpful in my understanding of the proposed approach.

Line 284-285: ... the scheme of angle calculation is the same as multi-view vision based method.

What does it mean? Could you elaborate?

Line 328-329: Spearman's correlation was used to evaluate the relative agreement between these methods.

Spearman assesses how well the relationship between two variables can be described using a monotonic
function. Do you assume that the relationship between these variables are monotonic? What is the
basis for this assumption?

Line 369: "show" --> "shown"

Line 378-379: Our results are comparable with previous study which uses a single Kinect v2 to capture
3D keypoints position of the subject.

Which previous study? Citation?

Line 382: "There are previously works" change to "There previous studies."

Line 449: It is an extra burden for CD patients with remote diagnosis and treatment at home.

Please elaborate with 2-3 sentences. Is it an extra burden because the cameras must be placed
at home at the patient's costs? Is it privacy?

Line 451: In the future work, more and more CD patients will gradually participate in this study.

Could you explicitly state how many patients participated in your pilot study?

Line 458: ... can describe in more special keypoints of humans.

This is not a grammatically correct sentence.

Section 6 on Line 465 should be renamed as "Summary."

Round 2

Reviewer 1 Report

The authors have addressed all concerns accordingly in the revision.